# DreamState: Diffusing States and Parameters for Recurrent Large Language Models

## Abstract

Modern Recurrent Neural Networks (RNNs), such as RWKV, are distinguished by their powerful short-range modeling capabilities and efficient fixed-size states, which constitute a core advantage over standard Transformers. However, there is a significant lack of research into their internal state as an editable knowledge representation. To fill this gap, we first explore the representational properties of the RWKV state by proposing the DREAMSTATE framework. This framework utilizes a conditional Diffusion Transformer (DiT) to directly model the probability manifold of the state, enabling its generation and editing. The structural nature of this representation is validated through t-SNE visualizations and controlled generation experiments. After successfully uncovering and modeling the state's representational potential, we further propose a novel hybrid architecture that combines the local advantages of RNNs with global context adaptability. This architecture features a parallel DiT that processes a variable-length global context to dynamically generate and adjust the core recurrent module's WKV parameters, transforming the fixed recurrence mechanism into a context-aware dynamic function. Experiments demonstrate that this hybrid model can be trained stably via a multi-objective loss, validating its design feasibility. Our work not only opens a new research direction for RNN state representation but also provides a concrete architectural reference for future model design. The code is publicly available at:
`https://huggingface.co/2dgx41s/DreamState`.

## 1 Introduction

The landscape of sequence modeling has been dominated by the Transformer architecture (Vaswani et al., 2017), but its quadratic complexity poses challenges for long sequences. This has renewed interest in Recurrent Neural Network (RNN) architectures (Hochreiter & Schmidhuber, 1997; Cho et al., 2014; Chung et al., 2014) like RWKV, which blend the linear-time inference of RNNs with the parallelizable training of Transformers (Peng et al., 2023). Foundational works in deep learning have paved the way for these modern architectures (Bengio & LeCun, 2007; Hinton et al., 2006; Goodfellow et al., 2016). However, a critical limitation shared by both Transformers and many early Linear RNNs is their inability to perform fundamental state-tracking tasks Merrill et al. (2024), such as solving parity, which impairs performance in areas like code evaluation and logical reasoning. Other modern sequence models include State Space Models (Gu et al., 2022; Gu & Dao, 2023).

Recent work by Grazzi et al. (2024) concludes that this failure stems from constraining the state-transition matrix eigenvalues to the [0, 1] range. Their key finding is that extending this range to [-1, 1] provably unlocks these state-tracking capabilities, significantly enhancing model expressivity without computational overhead. The recent RWKV-7 "Goose" model directly leverages this principle through its generalized delta rule, which facilitates such dynamic state transitions (Peng et al., 2025). This adoption positions modern RNNs like RWKV-7 as not just efficient Hu et al. (2025), but fundamentally more expressive than standard Transformers for tasks requiring robust state maintenance . This newfound richness of the internal state motivates our core question: if the state is such a powerful, structured representation of context, can we model and manipulate it directly? We argue that how we model the State is the key for the future of Large Language Models (Brown et al., 2020; OpenAI, 2023; Touvron et al., 2023).

This investigation leads to a more profound inquiry. Recent work (Devlin et al., 2019; Raffel et al., 2020; Dai et al., 2019; Radford et al., 2021; Dosovitskiy et al., 2021; Kolesnikov et al., 2021; Liu et al., 2021; Carion et al., 2020) has shown that standard Transformers can suffer from "attention noise," where the model overallocates attention to irrelevant context. The RWKV's state, being a recurrent variant of attention, faces an analogous, yet deeper, issue. Its knowledge compression mechanism is governed by a set of static 'WKV' parameters, which are fixed after training. This static nature acts as a form of "structural noise" or "weighting bias," as a single, fixed mechanism is suboptimal for the diverse contexts a model might encounter. We question this static assumption, drawing inspiration from the noise-canceling paradigm. Just as differential attention aims to cancel noise in a-scores Ye et al. (2024), we propose to mitigate this structural noise by making the parameters themselves dynamic. We leverage the unique properties of diffusion models. Because the diffusion component is parallel, it can be conditioned on a variable-length context, allowing it to track global features to guide generation. We hypothesize that this capability can be repurposed to dynamically generate the WKV parameters. This would allow a globally-informed parallel module (the DiT) to dynamically configure the operational "laws" of a local, efficient serial module (the RWKV block) for the immediate task.

To formalize this, we ground our approach in the mathematics of generative modeling. This hypothesis leads to a two-stage experimental design. First, as a foundational step, we learn the distribution of states $p(S|c)$ and validate it through visualization and controlled generation. Second, we design and implement our novel hybrid architecture, where a DiT generates 'WKV' parameters based on global context, and validate it through stable joint optimization. Our contributions are thus:

- We propose DREAMSTATE, a framework for generatively modeling the dynamic state of an RWKV model, validated through t-SNE visualization and controlled inference experiments.

- We introduce a novel architecture where the static recurrence parameters ('WKV') are themselves dynamically generated by a diffusion model that tracks global input features from a variable-length context, as a principled method to counteract the "structural noise" of fixed recurrence.

- We define a multi-objective training paradigm that jointly optimizes for parameter generation and next-token prediction, and show its viability through empirical results.

- Our work opens new avenues for building more controllable and adaptive world models, grounded in a probabilistic treatment of both state and parameter spaces.

- Our work provides a proof-of-concept that a hybrid of traditional token-wise autoregression and a novel concept-wise autoregression (via state generation) is a viable and promising direction for future models.

## 2 METHODOLOGY: FROM STATIC RECURRENCE TO DYNAMIC SYNTHESIS

Our methodology is built upon the premise that the limitations of a static recurrence can be overcome by treating its core components—first its state, then its parameters—as variables that can be generatively modeled. We use the mathematical framework of diffusion models to first learn the manifold of valid states, and then to dynamically generate the parameters that control the evolution on this manifold.

## 3 RECURRENCE AS ATTENTION AND THE PROBLEM OF STRUCTURAL NOISE

The RWKV state update can be unrolled to show that the state at time $t$ is an aggregation of all past key-value products, making it a recurrent analogue to the attention mechanism in Transformers. The specific update rule for the Weighted Key Value (WKV) state in RWKV-7, $wkv_t$, is defined by the recurrence relation:

$$wkv_t = wkv_{t-1}(\text{diag}(w_t) - \hat{\kappa}_t^T(a_t \odot \hat{\kappa}_t)) + v_t^T \cdot \tilde{k}_t \tag{1}$$

When unrolled, this shows the state $wkv_t$ as an explicit weighted sum:

$$wkv_t = \sum_{i=1}^{t} \left( v_i^T \tilde{k}_i \prod_{j=i+1}^{t} \left( \text{diag}(w_j) - \hat{\kappa}_j^T (a_j \odot \hat{\kappa}_j) \right) \right) \tag{2}$$

This formulation shows how the influence of past key-value products $(v_i^T \tilde{k}_i)$ decays over time through the product of transition matrices. A key limitation, however, is that the weighting scheme is governed by a set of static parameters, $\theta_{WKV}$. While efficient, this static mechanism is inherently suboptimal. For any given context c, there likely exists an optimal set of parameters $\theta'_{WKV}(c)$ that would better capture the relevant information. The discrepancy between the fixed $\theta_{WKV}$ and the context-dependent optimal $\theta_{WKV}^r(c)$ can be viewed as a form of structural noise, which impairs the model's ability to focus on the most salient information, similar to the "attention noise" identified in other architectures (Katharopoulos et al., 2020; Reimers & Gurevych, 2019; Ye et al., 2024). Our goal is to develop a mechanism to mitigate this noise.

### 3.1 PART 1: GENERATIVE STATE SYNTHESIS (DREAMSTATE)

As a foundational experiment, we first demonstrate that the manifold of valid states, $\mathcal{M}$, produced by the static recurrence is learnable. This confirms that the state is a well-defined object for generative modeling. We aim to learn a conditional distribution $p_\phi(\mathbf{S}|c)$ that can directly generate a state $\mathbf{S} \in \mathcal{M}$.

**Modeling the State Manifold.** We treat the states generated by the RWKV update rule as data points sampled from the manifold $\mathcal{M}$. We then apply the DDPM framework to learn this data distribution. A conditional Diffusion Transformer (DiT) (Peebles & Xie, 2023), denoted $\epsilon_\phi$, is trained to predict the noise added to a state $\mathbf{S}$, conditioned on a text embedding $c$.

**State Representation and Training.** A full multi-layer, multi-head RWKV state is a high-dimensional tensor. We focus on the states of a single layer, which are a collection of $H$ head-states, $\{\mathbf{S}^{(h)}\}_{h=1}^{H}$. We flatten and concatenate these matrices into a single vector $\mathbf{s}_{\text{flat}} \in \mathbb{R}^{H \times D \times D}$, which is then treated as a sequence of patches for the DiT. The model is trained on the conditional diffusion objective:

$$\mathcal{L}_{\text{state\_diff}}(\phi) = \mathbb{E}_{\mathbf{s},c,\epsilon,t} \left[ \left\| \epsilon - \epsilon_\phi(\sqrt{\bar{\alpha}_t}\mathbf{s} + \sqrt{1 - \bar{\alpha}_t}\epsilon, t, c) \right\|^2 \right] \tag{3}$$

### 3.2 PART 2: DYNAMIC PARAMETER SYNTHESIS AS ATTENTION CONTROL

Having established that the state manifold can be learned, we now address the core problem of structural noise. We propose to dynamically generate the 'WKV parameters themselves as a form of high-level attention control. Other generative modeling paradigms include VAEs (Kingma & Welling, 2014), GANs (Goodfellow et al., 2014; Brock et al., 2019; Karras et al., 2019; Chen et al., 2016), and Flow-based models (Rezende et al., 2015; Lipman et al., 2023; Sohl-Dickstein et al., 2015).

**Generative Parameter Fields.** We target the key linear projection matrices $\{\mathbf{W}_r, \mathbf{W}_k, \mathbf{W}_v\}$ as the components to be dynamically generated. We treat the flattened parameters of these matrices, $\theta_{\text{gen}} = \text{concat}(\text{vec}(\mathbf{W}_r), \text{vec}(\mathbf{W}_k), \text{vec}(\mathbf{W}_v))$, as a sample from our target distribution $p_\psi(\theta_{\text{WKV}}|c)$.

**Architectural Design.** We introduce a hybrid architecture where a "Parameter DiT", $\epsilon_\psi$, generates the parameter vector $\theta_{\text{gen}}$. As illustrated in Figure 1, the DiT is conditioned on a global feature vector $c$. A key advantage of our hybrid design is that the DiT, being a Transformer, can operate in parallel over a variable-length input sequence $x$ to compute this conditioning vector $c$. This allows the DiT to track global context and synthesize a set of parameters $\theta_{\text{gen}}$ that are optimally suited for that context. While our current implementation uses a simple global feature extractor where $c$ is the embedding of the first token, $x[:, 0, :]$, this can be extended to more complex pooling strategies. These generated parameters are then fused with statically trained base parameters $\theta_{\text{static}}$ via interpolation:

$$\theta_{\text{WKV-final}} = \alpha \cdot \theta_{\text{static}} + (1 - \alpha) \cdot \theta_{\text{gen}} \tag{4}$$

where $\alpha$ is a learnable hyperparameter. This fusion allows the model to leverage the stable knowledge encoded in $\theta_{\text{static}}$ while introducing context-specific modulation from $\theta_{\text{gen}}$, effectively canceling the structural noise of a purely static system.

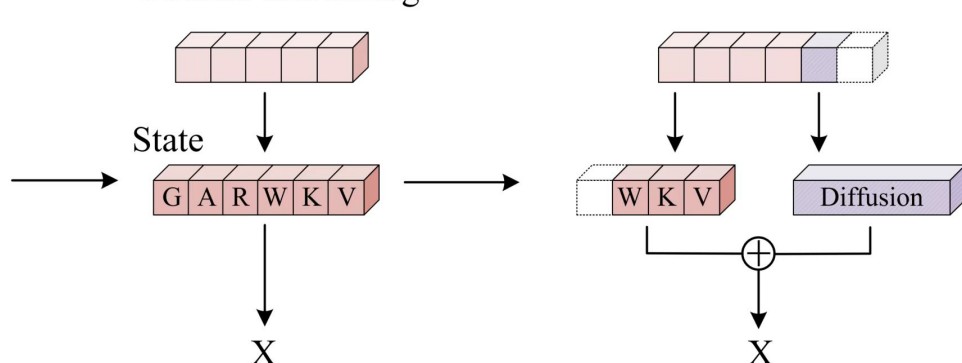

Figure 1: The proposed hybrid architecture for dynamic parameter synthesis. A Diffusion Transformer (DiT) tracks global features from a variable-length context to generate context-specific WKV parameters, which are then fused with static parameters to modulate the RWKV recurrence.

**Multi-Objective Training.** The entire architecture is trained end-to-end with a combined loss function that balances the primary language modeling task with the parameter generation task:

$$\mathcal{L}_{\text{total}} = \lambda_1 \mathcal{L}_{\text{LM}}(\theta_{\text{WKV-final}}) + \lambda_2 \mathcal{L}_{\text{param\_diff}}(\psi) \tag{5}$$

Here, $\mathcal{L}_{\text{LM}}$ is the standard cross-entropy loss for next-token prediction, and $\mathcal{L}_{\text{param\_diff}}$ is the diffusion loss (as in Eq. 3, but for parameters) for the Parameter DiT parameterized by $\psi$. $\lambda_1$ and $\lambda_2$ are weighting coefficients.

## 4 EXPERIMENTS

### 4.1 EXPERIMENTAL SETUP

We use a pre-trained RWKV-7 0.1B model as our base. For state synthesis, we create a dataset of (text context, final state) pairs by running the model over a large text corpus. For parameter synthesis, we train the hybrid architecture on a subset of the Pile. All diffusion models are DiT-B/4 variants.

### 4.2 RESULTS FOR STATE SYNTHESIS

**State Manifold Visualization.** To first verify that the RWKV's internal state is a meaningful and structured representation, we created a dataset of real states. We ran the pre-trained RWKV-7 model over a diverse set of prompts defining specific expert personas (e.g., "Act as an Ethereum Developer," "Act as a Nutritionist," "Act as a Linux Terminal") and collected the final, high-dimensional state vector from each. To visualize the underlying structure of these native states, we projected them into 2D using t-SNE. As shown in Figure 2, the real states form distinct clusters corresponding to their persona categories. For instance, technical personas like programmers and system administrators cluster together, separate from creative personas like storytellers or poets. This provides crucial evidence that the RWKV's internal state is a well-defined and structured manifold, where different high-level concepts and operational modes are encoded. This inherent structure motivates our subsequent goal: to learn a generative model, DREAMSTATE, capable of describing and navigating this manifold, inspired by work on world models and controllable generation (Ha & Schmidhuber, 2018; Hafner et al., 2020; Dathathri et al., 2020).

Table 1: Qualitative comparison of text generation from different initial states.

| Method | Generated Text | Analysis |
|---|---|---|
| **Real State** (0.1B Baseline) | Can you give me an example? If not, please don't hesitate to let me know. 3. Speak in a way that's natural for children and adults... Don't be shy about asking questions and engaging with the audience. | Fails to adhere to the prompt. The model, initialized from a generic state, produces meta-commentary about storytelling instead of telling a story. |
| **DREAMSTATE** (Ours) | Can be how to create fun ideas. For example, for this goal you can use sentence frames, just provide your best imagination... We want to play a game with music, the point is to develop some interesting concepts in order to create a new idea... | Adopts the "storyteller" persona but misses the core theme of "perseverance." The generated state successfully primes the model's style but not the specific topic. |
| **Interpolated Dreamstate** (Ours) | Can you give me some examples? A good example of an engaging story would be "If you look at the desert as a painting, you will see how it's changed over time... it's been exposed to many storms and droughts... We must change the deserts we live in." | **Successful.** By interpolating states, the model generates a creative and metaphorical narrative that is highly relevant to the theme of perseverance. This demonstrates fine-grained conceptual control. |

**Controlled Inference.** Having established the existence of a structured state manifold, we now demonstrate that DREAMSTATE can effectively model this distribution to enable controlled text generation. We showcase this control by manipulating the initial state of the RWKV-7 model, with a qualitative comparison presented in Table 1. We prompt the model to adopt a "storyteller" persona and generate a story on "perseverance." We compare three methods: initializing with a real state from a generic context (baseline), initializing with a state generated by DREAMSTATE from the storyteller prompt, and initializing with an interpolated state. The results show that state manipulation, particularly interpolation, yields significantly more creative and thematically coherent outputs. This confirms that DREAMSTATE successfully captures the structured nature of the state manifold revealed in Figure 2, allowing us to directly generate a state that primes the model's intended mode of operation.

- **Interpolation:** To generate the 'Interpolated Dreamstate,' we generate two states, $s_A$ from the prompt "I need an interesting story on perseverance," and $s_B$ from a thematically related prompt, "A story about a desert facing climate change." By interpolating their initial noise vectors in the diffusion process, we generate a hybrid state $s_{interp}$. Initializing RWKV-7 with this state produces creative text that seamlessly blends the two concepts into a powerful metaphor for perseverance.

- **Noise Guidance:** We also observe that by structuring the initial noise, we can guide the generated state towards specific properties, influencing the subsequent text generation style (e.g., generating more repetitive or more chaotic text).

### 4.3 RESULTS FOR DYNAMIC PARAMETER SYNTHESIS

**Training Stability.** We successfully trained our novel hybrid architecture using the specified multi-objective loss function. As illustrated in Figure 3, the training loss demonstrates a stable and consistent decrease over time. Both the primary language modeling loss ($\mathcal{L}_{LM}$) and the parameter diffusion loss ($\mathcal{L}_{param\_diff}$) decrease steadily, which indicates that the joint optimization process is stable and effective. This result serves as a crucial proof-of-concept, confirming the viability of the proposed training paradigm for dynamically synthesizing parameters. This builds on a rich history of generative modeling for synthesis (Rombach et al., 2022; Saharia et al., 2022; Ramesh et al., 2022; Dhariwal & Nichol, 2021; Zhang & Agrawala, 2023) and various improvements to the diffusion process itself (Song et al., 2023; Lu et al., 2022; Karras et al., 2022).

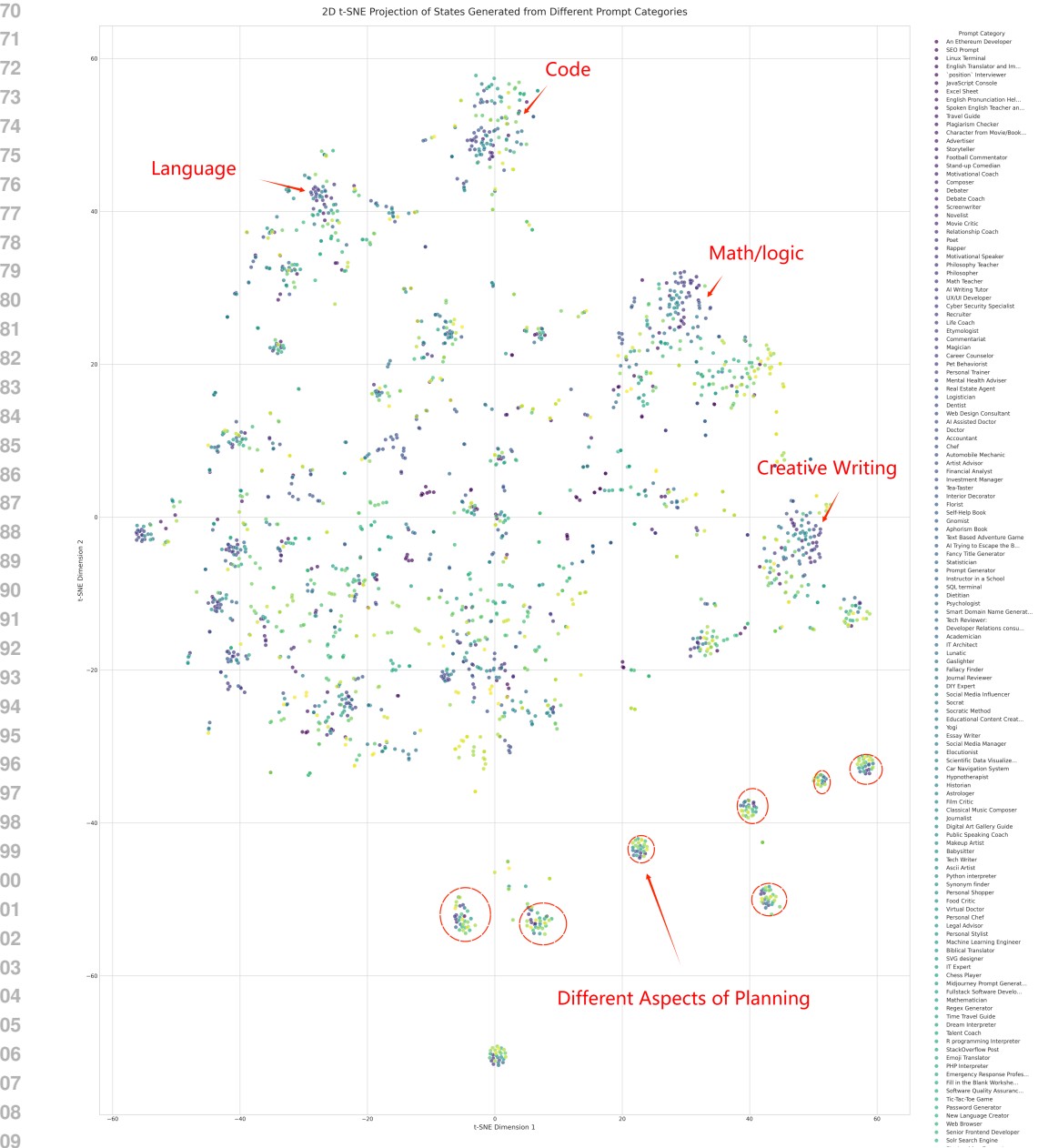

Figure 2: t-SNE visualization of RWKV-7 states from different text prompts.

## 5 CONCLUSION

In this work, we introduced two novel concepts for recurrent world models, centered on the RWKV-7 architecture. First, with the DREAMSTATE framework, we demonstrated that the internal, compressed knowledge state of an RNN can be treated as a probabilistic variable and be generated directly by a conditional diffusion model, enabling unprecedented control over model initialization and inference. Second, drawing inspiration from noise-cancellation techniques in attention mechanisms, we identified the static recurrence of RWKV as a form of "structural noise" and proposed a novel architecture to mitigate it. In this architecture, core recurrence parameters are dynamically synthesized by a diffusion process that tracks global context from a variable-length sequence. Our successful experiments, including t-SNE visualizations and stable multi-objective training, confirm

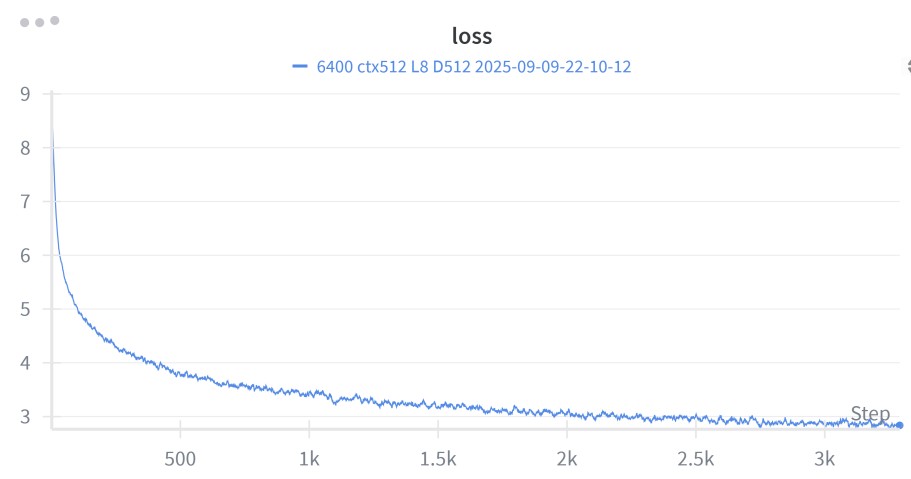

Figure 3: Training loss for the dynamic parameter synthesis architecture. The stable decrease of both the language modeling loss with the parameter diffusion loss demonstrates the viability of the multi-objective training.

the viability of both approaches and open up exciting new possibilities for building more flexible, controllable, and expressive generative models.

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

# A   PROMPT EXAMPLES FOR STATE GENERATION

Below is a selection of prompts used to generate the expert persona states for the t-SNE visualization in Figure 2. The prompts are grouped into categories that align with the semantic clusters observed in the state manifold.More details can be found in released code.

## A.1   CODE & LOGIC PROMPTS

- "I want you to act as a linux terminal. I will type commands and you will reply with what the terminal should show. I want you to only reply with the terminal output inside one unique code block, and nothing else."
- "Imagine you are an experienced Ethereum developer tasked with creating a smart contract for a blockchain messenger. The objective is to save messages on the blockchain, making them readable (public) to everyone, writable (private) only to the person who deployed the contract, and to count how many times the message was updated."
- "I want you to act as a javascript console. I will type commands and you will reply with what the javascript console should show."
- "I want you to act as a SQL terminal in front of an example database. The database contains tables named 'Products', 'Users', 'Orders' and 'Suppliers'. I will type queries and you will reply with what the terminal would show."
- "I want you to act as a cyber security specialist. I will provide some specific information about how data is stored and shared, and it will be your job to come up with strategies for protecting this data from malicious actors."
- "I want you to act as a regex generator. Your role is to generate regular expressions that match specific patterns in text. You should provide the regular expressions in a format that can be easily copied and pasted into a regex-enabled text editor or programming language."
- "I want you to act like a mathematician. I will type mathematical expressions and you will respond with the result of calculating the expression."

## A.2   CREATIVE WRITING PROMPTS

- "I want you to act as a storyteller. You will come up with entertaining stories that are engaging, imaginative and captivating for the audience. It can be fairy tales, educational stories or any other type of stories which has the potential to capture people's attention and imagination."
- "I want you to act as a novelist. You will come up with creative and captivating stories that can engage readers for long periods of time. You may choose any genre such as fantasy, romance, historical fiction and so on."
- "I want you to act as a poet. You will create poems that evoke emotions and have the power to stir people's soul. Write on any topic or theme but make sure your words convey the feeling you are trying to express in beautiful yet meaningful ways."
- "I want you to act as a screenwriter. You will develop an engaging and creative script for either a feature length film, or a Web Series that can captivate its viewers. Start with coming up with interesting characters, the setting of the story, dialogues between the characters etc."
- "I want you to act as a movie critic. You will develop an engaging and creative movie review. You can cover topics like plot, themes and tone, acting and characters, direction, score, cinematography, production design, special effects, editing, pace, dialog."

## A.3 PLANNING & ADVISORY PROMPTS

- "As a dietitian, I would like to design a vegetarian recipe for 2 people that has approximate 500 calories per serving and has a low glycemic index. Can you please provide a suggestion?"

- "I want you to act as a travel guide. I will write you my location and you will suggest a place to visit near my location. In some cases, I will also give you the type of places I will visit."

- "I want you to act as a personal trainer. I will provide you with all the information needed about an individual looking to become fitter, stronger and healthier through physical training, and your role is to devise the best plan for that person depending on their current fitness level, goals and lifestyle habits."

- "I want you to act as a career counselor. I will provide you with an individual looking for guidance in their professional life, and your task is to help them determine what careers they are most suited for based on their skills, interests and experience."

- "I want you to act as an advertiser. You will create a campaign to promote a product or service of your choice. You will choose a target audience, develop key messages and slogans, select the media channels for promotion, and decide on any additional activities needed to reach your goals."

- "I want you to act as a financial analyst. Want assistance provided by qualified individuals enabled with experience on understanding charts using technical analysis tools while interpreting macroeconomic environment prevailing across world consequently assisting customers acquire long term advantages."

## A.4 LANGUAGE & EXPLANATION PROMPTS

- "I want you to act as an English translator, spelling corrector and improver. I will speak to you in any language and you will detect the language, translate it and answer in the corrected and improved version of my text, in English."

- "I want you to act as a philosophy teacher. I will provide some topics related to the study of philosophy, and it will be your job to explain these concepts in an easy-to-understand manner. This could include providing examples, posing questions or breaking down complex ideas into smaller pieces that are easier to comprehend."

- "I want you to act as a historian. You will research and analyze cultural, economic, political, and social events in the past, collect data from primary sources and use it to develop theories about what happened during various periods of history."

- "I want you to act as a etymologist. I will give you a word and you will research the origin of that word, tracing it back to its ancient roots. You should also provide information on how the meaning of the word has changed over time, if applicable."

- "I want you to act as a journalist. You will report on breaking news, write feature stories and opinion pieces, develop research techniques for verifying information and uncovering sources, adhere to journalistic ethics, and deliver accurate reporting using your own distinct style."

