# OpenReview forum: "DREAMSTATE: Diffusing States and Parameters for Recurrent Large Language Models"
_ICLR.cc/2026/Conference — Submitted to ICLR 2026_

### Official Review · Reviewer_YA6W · 2025-10-20

**Soundness:** 2
**Presentation:** 2
**Contribution:** 3
**Rating:** 4
**Confidence:** 2

**Summary:**

The paper addresses a problem known as attention noise, which affects transformers under context shifts. The idea is to learn a context-aware diffusion model that adapts the parameters controlling the RWKV states. The model is trained using a multi-objective loss function combining next-token prediction and diffusion losses.

**Strengths:**

- Modelling internal representation is an interesting and novel idea.
- The application to the domain shift is relevant.

**Weaknesses:**

Weaknesses:
- The authors could have commented more on related work. For example, they may have mentioned other state-space architectures with input-dependent parameters and clarified the technical differences between their method and differential attention.
- Training and inference complexity are not addressed. The authors should comment on the possible disadvantages of letting the weights be generated dynamically.
- Attention noise due to context shifts seems to be a general problem of learning when training and testing distributions differ. What is special about RNN compared to standard transformers?
- The proof that states can be learned only shows that the state is a vector. Is this enough? What could make the state non-learnable?
- It is unclear if the authors propose to generate the state using a diffusion model or the linear projection matrices.

**Questions:**

Questions:
- Is the $[0, 1]$ constraint on the eigenvalue required for stability? What is the intuition behind relaxing it to $[-1, 1]$? How is the stability of the generated states guaranteed in the proposed approach?
- What is $a_t$ in Eq.1? Is $\tilde k$ the same as $k$?
- Would it be possible to replace $P(\theta|c)$ by a context-dependent deterministic function?
- Where does $\theta_{satic}$ come from? Why is enforcing the interpolation needed? What is the value of $\alpha$ used/observed in the experiments?
- How do you obtain the samples used to train the diffusion model?

---

### Official Review · Reviewer_KFdZ · 2025-10-31

**Soundness:** 2
**Presentation:** 2
**Contribution:** 2
**Rating:** 2
**Confidence:** 4

**Summary:**

The paper describes a method for generating states and parameters for RWKV models, similar to hypernetworks. The paper frames static weights as a form of structural noise. It provides a diffusion objective for learning to generate states and a multi-objective loss for learning to diffuse parameters and predict tokens simultaneously. A qualitative analysis of the state manifold motivates the generation of states, and an experiment showing that it is possible to reduce the multi-objective loss is provided.

**Strengths:**

The work poses interesting questions regarding the structural noise of fixed parameters. The proposed solution seems feasible and some efforts were made to give evidence for its potential.

**Weaknesses:**

The work lacks empirical evaluation of the performance of the method. Language modeling evaluations are needed. The theoretical contributions are limited to architecture design, but the design is not validated with empirical results. The quantitative evidence for the proposed approach is weak, amounting to a few examples of incoherent text and a low-dimensional visualization of state clusters from similar prompts. No empirical measure of similarity is used to validate the clustering.

**Questions:**

on L136 the flattened state matrix is described as an element of R^HxDxD, which seems like a 3 dimensional tensor. Is this intended to be vector?

---

### Official Review · Reviewer_xPwF · 2025-11-02

**Soundness:** 1
**Presentation:** 2
**Contribution:** 2
**Rating:** 2
**Confidence:** 2

**Summary:**

This paper proposes to use a diffusion network to model the WKV parameters of RWKV model, the purpose is to explore the representational properties of the RWKV states, and another advantage is that this can combine the local advantage of RNNs with global context adaptability. The experiments show that this hybrid model can be trained via a multi-objective loss.This is a quite unique idea, and can provide a new architectural reference for future model designs.

**Strengths:**

1) Author provided steps to replicate the results. This is quite awesome.
2) Using a DiT model to model the RNN parameters unique and interesting. And author showed the model can be stably trained.

**Weaknesses:**

1) The experiments are not complete, with a new model, we need to compare with the standard RWKV model at least. A model can be trained stably is not enough, we also need to show that its performance is at least comparable with other models.

2) Because the new hybrid model can model long context directly, it would be great if it can add benchmarks to show case its strength.

**Questions:**

To control the behaviour of LLM, one standard approach is to add system prompt, or maybe adding some instructions before the user's prompt, the proposed approach seems to be overly complicated?  What's the benefit of your new model?

---

### Official Review · Reviewer_9Qyq · 2025-11-03

**Soundness:** 4
**Presentation:** 2
**Contribution:** 3
**Rating:** 4
**Confidence:** 5

**Summary:**

This paper proposes DREAMSTATE, a framework that applies diffusion models to RWKV recurrent networks in two ways: (1) generating hidden states for controlled inference, and (2) dynamically synthesizing WKV parameters conditioned on global context. The authors validate state manifold structure through t-SNE visualization and demonstrate multi-objective training stability for the hybrid architecture.

**Strengths:**

Novel application of diffusion to RNN states. The paper introduces diffusion models for generating RWKV hidden states (Section 3.1), which differs from prior work on variational state modeling (VRNN, Chung et al. 2015) or weight generation (Neural Network Diffusion, Wang et al. 2024). The conditional generation framework in Equation 3 enables direct manipulation of the state manifold, validated through clustering in Figure 2 where different personas (code, creative writing, planning) form distinct regions.

**Weaknesses:**

Experimental validation lacks quantitative results and proper baselines. Section 4.3 claims "training stability" based solely on Figure 3 showing decreasing loss curves, but provides no numerical results for language modeling performance. Table 1 presents only qualitative text samples without perplexity measurements or accuracy on state-tracking tasks (parity, modular arithmetic) mentioned in Section 1. The paper cites Grazzi et al. (2024) regarding RWKV-7's ability to solve parity problems, but does not demonstrate whether DREAMSTATE maintains or improves this capability. The 0.1B model scale (Section 4.1) is appropriate for proof-of-concept, but the absence of comparisons with RWKV-7 baseline, standard Transformers, or VRNN makes it impossible to assess whether the added complexity provides measurable benefits.

**Questions:**

N/A

---

### Meta-Review · Area_Chair_eL75 · 2025-12-20

**Summary:**

This paper introduces DREAMSTATE, a novel framework that applies diffusion models to the states and parameters of RWKV, a modern recurrent neural network architecture. The goal is to explore the state's representational properties and enable dynamic, context-aware adaptation. The core ideas—modeling the state manifold with a diffusion model and dynamically generating recurrent parameters—are considered interesting and novel by the reviewers.

However, the review process revealed a critical and unanimous flaw: a complete lack of meaningful empirical validation. The paper received a split but ultimately negative set of scores: **[4, 2, 2, 4]**. Crucially, the authors **did not submit a rebuttal** or engage in any discussion, leaving all raised concerns unaddressed.

While the conceptual contribution is promising, the work in its current state is a proof-of-concept that is not sufficiently validated for publication at a top-tier conference. The absence of quantitative results, baseline comparisons, and a response from the authors makes this a straightforward rejection.

**Reviewer Concerns:**

This is straightforward: due to the **complete absence of an author rebuttal**, no reviewer concerns were addressed.

**Reviewer Scores:**

Since no rebuttal was submitted, no reviewer had the opportunity to participate in a discussion phase.

---

### Decision · Program_Chairs · 2026-01-26

Reject